# Nysfungin Production Improvement by UV Mutagenesis in *Streptomyces noursei* D-3-14

Ming Song [1], Wubing He [2], Sulan Cai [1], Fuju Wang [3], Weizhuo Xu [2],* and Wei Xu [2],*

1   School of Functional Food and Wine, Shenyang Pharmaceutical University, 103 Wenhua Road, Shenhe District, Shenyang 110016, China
2   School of Life Sciences and Biopharmaceuticals, Shenyang Pharmaceutical University, 103 Wenhua Road, Shenhe District, Shenyang 110016, China
3   Beijing Global Biotechnologies, Co. Ltd., No.99 Yuexiu Road, Haidian District, Beijing 100193, China
*   Correspondence: weizhuo.xu@syphu.edu.cn (W.X.); shxuwei8720@163.com (W.X.);
    Tel.: +86-024-43520301 (W.X.); +86-024-43520307 (W.X.); Fax: +86-024-43520301 (W.X.);
    +86-024-43520307 (W.X.)

**Abstract:** *Streptomyces noursei* D-3-14 was taken as a starting strain and treated with UV (15 W, 30 cm) mutagenesis for 40 s for three consecutive rounds. High yielding strains were screened using chemical and biological potency determination, and the components of the fermentation products were detected using HPLC. Finally, the mutant strain *Streptomyces noursei* 72-22-1 with a chemical potency of 8912 (U/mL) and a biological potency of 5557 (U/mL) was obtained after the genetic stability evaluation. After optimization of the fermentation conditions, the chemical potency and biological potency of *Streptomyces noursei* 72-22-1 reached 14,082 U/mL and 10579 U/mL, respectively, which is 1.58 and 1.91 times those before optimization. HPLC analysis indicated that the mutant strain 72-22-1 displayed a higher content of polyfungin B. When equimolar nystatin A1, A3, and polyfungin B were tested for their fungicidal activities towards *Saccharomyces cerevisiae* ATCC 2061, polyfungin B exhibited a better efficacy than nystatin A1 and A3.

**Keywords:** *Streptomyces noursei*; nysfungin; nystatin A1; nystatin A3; polyfungin B; UV mutagenesis





## 1. Introduction

Nystatin is a kind of polyene macrolide antibiotic [1–3] that was first isolated by Hazen and Brown from soil in Fauquier County, Virginia in the 1950s [4]. At that time, it was named Fungicidin and demonstrated fungistatic and fungicidal activities, without antibacterial action. In 1954, this Fungicidin was first commercialized by Bristol Myers Squibb and renamed nystatin, with its major ingredient being nystatin A1. Later, Chinese researchers also isolated and identified additional ingredients of actinomycetes derived from Fungicidin in Guangdong province soil [5,6]. In 1981, Thomas et al. compared pharmaceutical grade samples of Fungicidin from China, Hungary, Italy, US, and Russia, and identified that nystatin A1 represented the majority of the ingredients in the US, Italy, and Hungarian samples, with a 70% concentration [7], which coincides with the British Pharmacopoeia in 1980 [8]. Meanwhile, he also reported that only 12% nystatin A1 was founded in the Chinese samples, with other components nystatin A3 and polyfungin B ranging from 20 to 50% [9]. Since then, nystatin has usually been assigned as western nystatin, which has a high A1 content as the major ingredient. While nysfungin is normally used to name the Chinese-derived nystatin, besides the A1, which may also have A3 and polyfungin B.

Nystatin is effective against *Candida*, *Cryptococcus*, *Aspergillus*, *Histoplasma*, and *Blastomyces* [10,11]. Nystatin is one of the most commonly used topical antifungal drugs, with a high efficacy, low cost, and fewer side effects, due to not being absorbed from the gastrointestinal tract [12,13].

From their chemical structures, it could be seen that nystatin A3 has an additional digitoxose compared to nystatin A1 [9,14], while the polyfungin B has an absence of a C10 hydroxyl group compared with nystatin A3 [15]. All three components share the same macrolide polyene skeleton, as shown in Figure 1.

**Figure 1.** Chemical structures of nystatin A1, A3, and polyfungin B.

In the FDA approved nystatin drugs, the major ingredient is the single nystatin A1, with the CAS No. 34786-70-4. While in the Chinese pharmacopoeia [16], nystatin is still considered a multiple-ingredient drug, and was named Nysfungin, so as to differentiate it from the single major A1 nystatin.

Due to the actinomyces metabolic diversity, in this research, we used an environmentally isolated strain of *Streptomyces noursei* D-3-14 to perform ultraviolet (UV) mutagenesis, optimized the culture conditions, and finally obtained a higher production of Nysfungin. Mutation breeding is a powerful technique, in which microbial strains exposed to mutagen treatment are screened, to identify positive mutants with specific characteristics. This method has the advantages of simplicity, rapidity, and high efficiency [17]. UV is possibly one of the physical mutagens that causes genetic variation and enables the selection of traits as needed. UV has a strong genotoxic effect, producing DNA damage that leads to an altered DNA structure [18]. UV irradiation induces covalent crosslinks between neighboring pyrimidines. If left unrepaired, error-prone replication of this damaged DNA leads to an increased rate of mutagenesis and genome instability [19]. The aim of this study was to screen and characterize nystatin-producing *Streptomyces noursei* strains exposed to UV, to identify the strain that produced the highest activity of nystatin, thereby obtaining a high-activity and low-cost raw material for commercial nystatin production.

## 2. Results and Discussion

### 2.1. Determination of the Chemical and Biological Potency of the Starting Strain Streptomyces noursei D-3-14

According to the experimental process in Section 3.3, the chemical potency of the starting strain *Streptomyces noursei* D-3-14 was detected as 3464 U/mL, as shown in Table 1, and the biological potency was detected as 2703 U/mL, as shown in Table 2.

**Table 1.** Chemical potency detection of *Streptomyces noursei* D-3-14.

| Expt. No | OD$_{319}$ | Chemical Potency (U/mL) | Average Chemical Potency (U/mL) |
|----------|------------|-------------------------|----------------------------------|
| 1 | 0.295 | 3709 | |
| 2 | 0.249 | 3116 | 3464 |
| 3 | 0.284 | 3567 | |

**Table 2.** Biological potency detection of *Streptomyces noursei* D-3-14.

| Expt. No | Inhibition Diameter (mm) 80 U/mL Reference | Inhibition Diameter (mm) 40 U/mL Reference | Inhibition Diameter (mm) 80 U/mL Sample | Inhibition Diameter (mm) 40 U/mL Sample | Biological Potency (U/mL) | Average Biological Potency (U/mL) |
|---|---|---|---|---|---|---|
| 1 | 23.42 | 19.00 | 21.64 | 17.50 | 2658 | |
| 2 | 21.98 | 18.44 | 20.26 | 17.18 | 2538 | 2703 |
| 3 | 23.22 | 18.94 | 22.16 | 17.88 | 2914 | |

*2.2. HPLC Analysis of the Starting Strain Streptomyces noursei D-3-14*

According to the experimental process in Section 3.3, the HPLC analysis of D-3-14 was similar to the nysfungin reference, as seen in Figure 2.

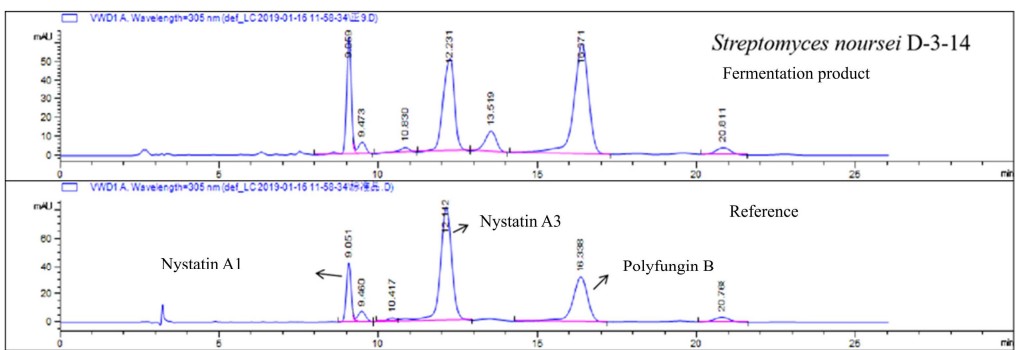

**Figure 2.** HPLC analysis of the fermentation products of *Streptomyces noursei* D-3-14 and the reference.

*2.3. Determination of the UV Irradiation Duration of Mutagenesis*

The lethal rate should increase with UV radiation duration. According to our experience, a 90% lethal rate is acceptable for further research. Figure 3 shows that the UV radiation duration was set to 40 s.

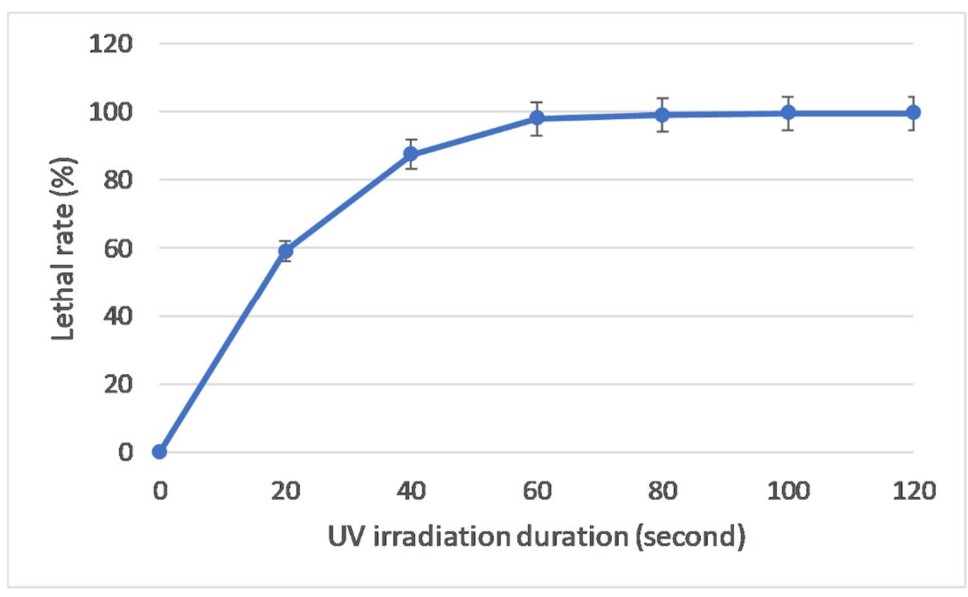

**Figure 3.** Lethality curve for UV irradiation duration.

*2.4. Preliminary and Secondary Screening for Three Consecutive Rounds*

According to the experimental process in Section 3.4, three consecutive rounds of preliminary and secondary screening were performed following the UV mutagenesis.



In the first round of preliminary screening, 74 mutants were obtained. Then 14 out of these 74 mutants were selected for the secondary screen. In this round, *Streptomyces noursei* 74-14-8 was identified as the resultant strain, with a chemical potency of 5783 U/mL and biological potency of 4626 U/mL, which exhibited about 1.67 and 1.71 times the chemical and biological potency of the starting strain, as seen in Table 3.

**Table 3.** Biological potency and Polyfungin B contents of various strains after the first round of UV mutagenesis.

| Strain No. | Chemical Potency (U/mL) | Biological Potency (U/mL) | Polyfugin B Content (%) |
|---|---|---|---|
| 74-14-8 | 5783 | 4626 | 33.63 |
| 74-14-61 | 5177 | 3467 | 27.64 |
| 74-14-67 | 5101 | 3825 | 21.61 |

In the second round of screening, 22 out of 72 preliminary screened mutants were selected. The resultant *Streptomyces noursei* 72-22-1 was identified as having a chemical and biological potency of 8912 U/mL and 5557 U/mL, exhibiting 1.54 and 1.20 times the chemical and biological potency of *Streptomyces noursei* 74-14-8 (Table 4).

**Table 4.** Biological potency and Polyfungin B contents of various strains after the second round of UV mutagenesis.

| Strain No. | Chemical Potency (U/mL) | Biological Potency (U/mL) | Polyfugin B Content (%) |
|---|---|---|---|
| 72-22-1 | 8912 | 5557 | 53.63 |
| 72-22-3 | 5328 | 3467 | 47.64 |
| 72-22-5 | 7214 | 5679 | 31.61 |
| 72-22-10 | 5912 | 3578 | 44.72 |
| 72-22-11 | 6879 | 4344 | 35.11 |
| 72-22-14 | 6020 | 4684 | 36.79 |
| 72-22-19 | 5829 | 3689 | 32.77 |
| 72-22-20 | 5741 | 4121 | 33.04 |
| 72-22-49 | 5311 | 3877 | 28.64 |

In the third round of screening, 24 out of 112 preliminary screened mutants were selected. The resultant *Streptomyces noursei* 112-24-63 was identified as having a chemical and biological potency of 11097 U/mL and 10751 U/mL, which was 1.25 and 1.93 times the chemical and biological potency of *Streptomyces noursei* 72-22-1 (Table 5).

**Table 5.** Biological potency and Polyfungin B contents of various strains after the third round of UV mutagenesis.

| Strain No. | Chemical Potency (U/mL) | Biological Potency (U/mL) | Polyfugin B Content (%) |
|---|---|---|---|
| 112-24-11 | 8692 | 6945 | 39.66 |
| 112-24-23 | 4463 | 5007 | 27.46 |
| 112-24-38 | 4353 | 5010 | 26.10 |
| 112-24-63 | 11,097 | 10,751 | 31.86 |
| 112-24-70 | 7837 | 9044 | 27.92 |
| 112-24-78 | 5545 | 8539 | 22.14 |
| 112-24-97 | 7340 | 2951 | 23.37 |

*2.5. Genetic Stability Experiment*

According to the experimental process in 3.5, genetic stability experiments were performed on *Streptomyces noursei* 72-22-1 and 112-24-63. The results can be seen in Figure 4.

It was observed that Strain 112-24-63 could not withstand the stability test, but 72-22-1 could endure five consecutive generations of culture. Later, *Streptomyces noursei* 72-22-1 underwent fermentation optimization for a better potency.

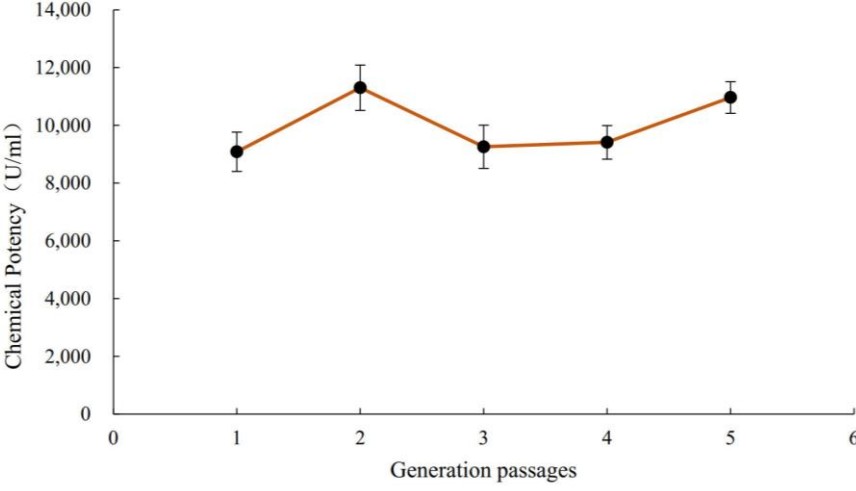

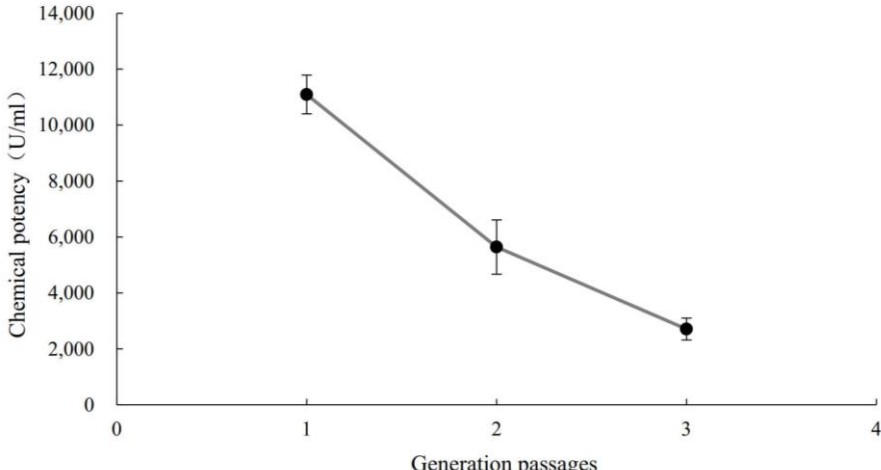

**Figure 4.** Genetic stability experiment results for *Streptomyces noursei* 72-22-1 and 112-24-63.

### 2.6. Single-Factor Evaluation for the Nysfungin Fermentation

Single factor glucose concentration, peanut meal concentration, starting pH, and inoculation volume ration were evaluated to generate the results in Figure 5. The results showed that when the content of glucose in the fermentation medium was 5.5%, the yield of nystatin was higher. A carbon source is one of the essential nutrients in a microbial culture medium, providing energy for the growth, reproduction, and metabolic activities of microorganisms. Our previous work found that during streptomycin fermentation, the glucose concentration in the fermentation broth must be controlled to be lower than a certain level in the later stage of fermentation. If the glucose concentration is higher than 10 mg/mL, the synthesis of mannosidase will be inhibited and the production of streptomycin will be significantly reduced. When the culture medium contains two or more carbon sources, microorganisms generally use glucose first and then the other carbon sources. These results indicate that secondary metabolites, such as antibiotics, are regulated by carbon metabolites, such as glucose.

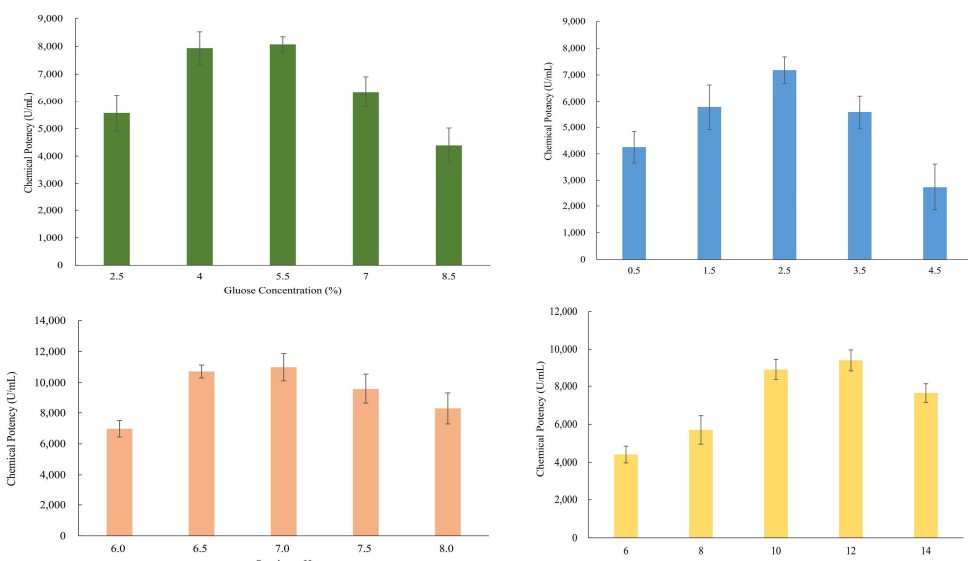

**Figure 5.** Evaluation results for single factors of glucose concentration, peanut meal concentration, starting pH, and inoculation volume ratio for fermentation.

The nitrogen source not only plays a nutritional role in the fermentation of microorganisms, but also contains inducers, precursors, and other substances required for the synthesis of secondary metabolites. When there are multiple nitrogen sources in the fermentation medium, microorganisms always use the simple nitrogen sources first, and then decompose complex nitrogen sources. Moreover, when the concentration of these simple nitrogen sources (such as ammonium ions, amino acids) is high, they synthesize few secondary metabolites. The results showed that when the content of peanut meal in the fermentation medium was 2.5%, the yield of nystatin was higher.

The experimental results showed that when the initial pH of the fermentation medium was 7.0, and the inoculation volume was 12%, the yield of nystatin fermentation was higher.

According to the above single-factor evaluation results, a four-factor and three-level $L_9$ ($3^4$) orthogonal experiment was designed, as seen in Table S1. Orthogonal experimental results and analysis (Table S2) showed that a glucose concentration of 5.5%, peanut melt concentration of 2.5%, initial pH of 7.5, and inoculation volume ratio of 14% were the optimal conditions. The final optimized chemical and biological potency results for *Streptomyces noursei* 72-22-1 were 14,082 U/mL and 10579 U/mL, which are 1.58 and 1.91 times those before optimization.

### 2.7. HPLC Analysis for Streptomyces noursei 72-22-1

According to the experimental process in 3.3, the fermentation product in the HPLC analysis of 72-22-1 was similar to the nysfungin reference, as seen in Figure 6.

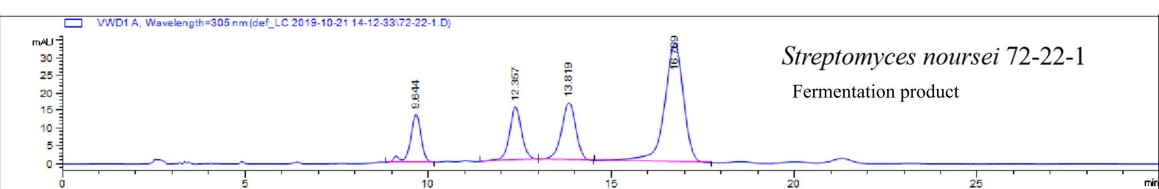

**Figure 6.** HPLC analysis of the fermentation products of *Streptomyces noursei* 72-22-1.

It can be seen that in the mutant *Streptomyces noursei* 72-22-1, nystatin A1, A3, and polyfungin B exhibited retention times of 9.085 min, 12.288 min, and 16.599 min. The ratio of polyfungin B was about 53%, which is much higher than the reference.

*2.8. Fungicidal Activity of Nystatin A1, A3, and Polyfungin B towards Saccharomyces cerevisiae ATCC 2061*

Single ingredients of nystatin A1, A3, and polyfungin B were extracted and isolated using semi-preparative HPLC, and their fungicidal activity was evaluated according to the experimental process in 3.3.

The results are summarized in Table 6. It can be seen that polyfungin B displayed a better fungicidal activity than nystatin A1 and A3, which is a novel result in nysfungin activity research.

**Table 6.** Results of nystatin A1, A3, and polyfungin B inhibiting *Saccharomyces cerevisiae* ATCC 2061.

| Expt. No. | Inhibition Zone Diameter of Nystatin A1 (mm) | Inhibition Zone Diameter of Nystatin A3 (mm) | Inhibition Zone Diameter of Polyfugin B (mm) |
|---|---|---|---|
| 1 | 20.68 | 16.44 | 21.62 |
| 2 | 19.26 | 16.84 | 21.18 |
| 3 | 18.56 | 15.84 | 19.12 |
| 4 | 19.76 | 16.60 | 20.18 |
| 5 | 19.14 | 16.52 | 20.08 |
| Average Value | $19.48 \pm 0.71$ | $16.45 \pm 0.33$ | $20.44 \pm 0.88$ ** |

** $p < 0.05$, highly significant.

## 3. Materials and Methods

### 3.1. Chemicals, Strains, Culture Media, and Growth Conditions

Due to the lack of a nysfungin standard, domestic commercial nysfungin tablets (Nysfungin, Zhenyuan Pharmaceuticals Co. Ltd., Shaoxing City, Zhejiang, China) were purchased and processed to be used as a reference. All other chemicals were analytically pure, unless expressed otherwise.

The starting *streptomyces noursei* D-3-14 was isolated and stored in our lab and had a chemical potency of 3464 U/mL and a biotic potency of 2703 U/mL. Strains were cultured at 28 °C for 7 days on Gause's synthetic solid medium slant (GA media). A loop of the freshly grown lawn was inoculated into 50.0 mL seed culture medium, and cultured at 150 rpm, 28 °C for 24 h. Then, 10% of this seed culture was inoculated into 50 mL fermentation culture medium and cultured at 200 rpm, 28 °C, for 86 h. The resultant culture could be used to extract nystatin.

Gause's synthetic solid medium consisted of (g/L) soluble starch, 20.0; $KNO_3$, 1.0; NaCl, 0.5; $K_2HPO_4$, 0.5; $MgSO_4 \cdot 7H_2O$, 0.5; $FeSO_4 \cdot 7H_2O$, 0.01 and agar, 15, pH 7.2. Seed culture medium consisted of (g/L) peanut meal, 20.0; soluble starch, 10; glucose, 10; $CaCO_3$, 6.0; soybean oil, 3.0; $(NH_4)_2SO_4$, 2.0; peptone 2.0; $MgSO_4 \cdot 7H_2O$, 0.5; $K_2HPO_4$, 0.2 and pH 7.2. Fermentation culture medium consisted of (g/L) glucose, 55.0; peanut meal, 25.0; $CaCO_3$, 10.0; $(NH_4)_2SO_4$, 3.0; peptone 3.0; dried silkworm chrysalis meal, 2.0; soybean oil, 1.0; $KH_2PO_4$, 0.02 and pH 7.2.

*Saccharomyces cerevisiae* ATCC 2061 was purchased from China General Microbiological Culture Collection Center (CGMCC), Beijing, China. It was cultured on a YM medium slant at 28 °C for 3 d. When was mature, 6.0 mL sterilized NaCl solution (0.9%) was added, and the lawn was scraped to obtain a suspension. The suspension was then transferred into Erlenmeyer flasks with glass beads and incubated 28 °C, 180 rpm for 30 min. The final suspension concentration was adjusted and calculated using a hemocytometer.

The YM culture medium consisted of (g/L) glucose, 10.0; peptone 5.0; yeast extract, 5.0; malt extract, 5.0; agar 15.0 and pH 6.2. The biological potency detection culture medium consisted of (g/L) maltose, 40.0; peptone 20.0; NaCl, 3.0; agar 15.0 and pH 7.0–7.2. The Gauss's culture medium was sterilized at 121 °C for 20 min, all other media were sterilized at 115 °C for 30 min.

### 3.2. UV Mutagenesis, Nysfungin Crude Extraction, and HPLC Detection

First, 3 mL of *Streptomyces noursei* single spore suspension was added to the 75 mm culture dishes and radiated with UV light (15 W, 30 cm) for 60 s. The dish covers were then opened and radiated for 20 s, 40 s, 60 s, 80 s, 100 s, and 120 s. Next, 12 mL seed culture medium was supplemented, and the dishes were cultured in black bags at 28 °C for 2 h. Both the UV radiated (Mutagenesis group) and untreated spore (Control group) suspensions were serially diluted and calculated using a hemocytometer. Then, 100 uL of spore suspension with $10^{-3}$, $10^{-4}$, and $10^{-5}$ concentrations were spread on the Gauss plates, each concentration was spread in three replicates, the plates were cultured at 28 °C for 5~7 days, the colony numbers were counted, and the lethal rate was calculated.

$$\text{Lethal rate (\%)} = \frac{(\text{cfu of control group} \times \text{dilution ratio}) - (\text{cfu of Mutagenesis group} \times \text{dilution ratio})}{(\text{cfu of control group} \times \text{dilution ratio})} \times 100\%$$

After 86 h, the fermentation broth was centrifugated at 3500 rpm for 10 min, to collect the wet mycelia. These wet mycelia were transferred into a brown beaker, 3 times the volume of 98% alcohol (*v/v*) was added and stirred for 40 min, then the solution was centrifugated at 3500 rpm for 10 min, the supernatant was collected and extracted with an equal volume of 98% alcohol (*v/v*) three times. All the supernatants were combined and vapored with reduced pressure to concentrate to 20 mL, and these solutions were kept at 4 °C overnight to obtain the nystatin suspension. These suspensions could be further processed at 4000 rpm for 6 min to eliminate the supernatant and to obtain the crude nystatin extract crystal and were stored at −20 °C before use.

Subsequently, 50.00 mg of the above crude nystatin crystal was dissolved in 5.00 mL methanol (chromatographic grade) and filtered using a 0.22 μm filter, to produce a 10 mg/mL solution. HPLC was performed using a WondaSil C18 Column (250 × 4.6 mm × 5 μm, Shimadzu, Kyoto, Japan), and the mobile phase consisted of methanol and acetonitrile, acetate buffer = 26:37:37, the detection wavelength was 305 nm, the column temperature was 30 °C, the loading speed was 1 ml/min, and the loading volume was 10 μL.

### 3.3. Determination of the Chemical and Biological Potencies

First, the nystatin reference was resolved in *N,N*-dimethylformamide to make a 1000 U/mL stock. Then, 0 mL, 0.2 mL, 0.4 mL, 0.6 mL, 0.8 mL, 1.0 mL, 1.2 mL, 1.4 mL, 1.6 mL, 1.8 mL, and 2.0 mL stock was dissolved and filled to 25.00 mL, and the absorption values were detected on OD319 to establish a standard curve.

Chemical potency was detected using the following process adapted from the European Pharmacopoeia 8.0 [20]: First, after 86 h, 5 mL well-dispersed culture broth was washed three times with ddH$_2$O and centrifugated to collect the mycelia. Then, the mycelia were well mixed with 10 mL methanol and left to stand for 2 h. Next, the supernatant was collected after 3500 rpm for 10 min, and extracted using 10 mL methanol twice. The combined methanol solution was filled to 50.00 mL. Then, 2.0 mL was used to detect OD319, and the chemical potency was evaluated by comparing with the above standard curve.

The biological potency was detected using the following process adapted from a USP monograph [21]. The first step was to evaluate the proper concentration of *Saccharomyces cerevisiae* ATCC 2061. Then, 1 mL of $10^{-1}$, $10^{-2}$, $10^{-3}$, $10^{-4}$, and $10^{-5}$ dilution suspension was added to 15 mL biological potency detection culture medium at 48 °C and mixed well. When the media become solid, one Oxford cup was placed on the center of the petri dish and 100 uL nystatin reference solution (80 U/mL) was supplemented and the cover was put on. Each concentration was performed in triple replicates and cultured at 28 °C for 14–18 h. The diameter of the inhibition zone was measured with a Vernier caliper. Empirically, a diameter of 18–22 mm is acceptable for biological potency detection.

Then, 250 uL of *Saccharomyces cerevisiae* ATCC 2061 was added to the 15 mL biological potency detection culture medium at 48 °C and mixed well. When the media became solid, four Oxford cups were symmetrically placed on the petri dish. Two cups on one diagonal

line were supplemented with 100 uL nystatin reference solution (80 U/mL and 40 U/mL), while the other two cups on the other diagonal line were supplemented with 100 uL nystatin sample solution (80 U/mL and 40 U/mL). After being cultured at 28 °C for 14–18 h. The diameter of the inhibition zone was measured with a Vernier caliper. And the biological potency of the crude nystatin extracts was calculated using the following formula:

$$\theta = log^{-1}\left( \frac{T_2 - S_2 + T_1 - S_1}{S_2 + T_2 - S_1 - T_1} \times I \right)$$

in which,

$S_1$ = inhibition zone diameter of the low-dose control solution
$S_2$ = inhibition zone diameter of the high-dose control solution
$T_1$ = inhibition zone diameter of the low-dose sample solution
$T_2$ = inhibition zone diameter of the high-dose sample solution
$I = lg$(high-dose concentration/low-dose concentration)

### 3.4. Preliminary Screen and Secondary Screen for Mutants

Preliminary screening was performed using the following process: Single colonies on the UV radiation plates were inoculated onto a fresh Gauss's synthetic media plate and cultured at 28 °C for 5–7 days, to obtain viable mutants. A $1 \times 1$ cm$^2$ lawn was scraped and transferred into 50 mL fermentation broth and cultured at 28 °C and 180 rpm for 86 h. Then, the crude nystatin was extracted, to examine the chemical potency. Those strains with higher chemical potency than the starting strains were screened as the preliminary strains.

A secondary screen was performed on those strains with higher chemical potencies. Those strains were further detected by HPLC to determine their biological potencies. Only those strains exhibiting both higher chemical and biological potencies were screened and selected for the next steps.

### 3.5. Genetic Stability Evaluation

The selected strains with high chemical and biological potency were cultured for five consecutive generations, and the chemical potency was tested for each generation to evaluate the genetic stability.

### 3.6. Single Ingredient Evaluation for the Obtained Nysfungin

The obtained Nysfungin was detected using HPLC, each of the major peaks was further extracted and isolated using semi-preparative HPLC, and the resultant single ingredients were identified by MS and NMR and tested for their fungicidal activities using the cup-plate methods in 3.3.

### 3.7. Optimization of the Fermentation Conditions

The culture process was performed as described in 3.1. The glucose concentration was set at 2.5%, 4.0%, 5.5%, 7.0%, and 8.5%. The peanut meal concentration was set at 0.5%, 1.5%, 2.5%, 3.5%, and 4.5%. The starting pH value was set at 6.0, 6.5, 7.0, 7.5, and 8.0. The inoculation volume ratio was set at 6%, 8%, 10%, 12%, and 14%. Each inoculation volume was replicated three times.

The orthogonal experiment with four factors and three levels $L_9$ ($3^4$) was designed according to the glucose concentration, peanut meal, starting pH, and inoculate volume ratio, determined in a single factor experiment.

### 3.8. Statistical Analysis

All experiments were carried out in triplicate and each presented value is the average of three independent experiments. SPSS 20.0 software was used to conduct *t*-tests on the data, to determine the statistical difference, $p < 0.05$ was significant (*), $p < 0.01$ was extremely significant (**).

## 4. Conclusions and Discussion

As polyene macrolide antibiotics produced by the *Streptomyces noursei strain*, both nystatin and nysfungin have broad-spectrum antifungal effects, with the strongest inhibition of *Candida albicans*, and have been widely used in clinical practice [3]. In this study, *Streptomyces noursei* D-3-14, a nystatin producing strain, was treated with UV mutagenesis for three rounds and screened for high yield mutants. Finally, *Streptomyces noursei* 72-22-1, a genetically stable strain with enhanced nystatin production, was obtained. Its chemical potency was 8912 U/mL and its biological potency was 5557 U/mL, 2.57 times and 2.06 times of those of the original strain, respectively. After optimizing the fermentation conditions, the chemical potency and biological potency of *Streptomyces noursei* 72-22-1 were 14,082 U/mL and 10579 U/mL, respectively, 4.07 and 3.92 times those of the starting strain.

Admittedly, although antibiotics have been applied for many years, there are only a few clinical options for polyene macrolide ingredients. In the FDA Orange Book dated December 2022, there are only four polyene macrolide fungicidal antibiotics listed. The only discontinued ingredient is Candicidin (Vanobid), which was approved on 1 January 1982 and manufactured by Sanofi Aventis US. The Eyevance Pharmeceuticals-manufactured Natamycin (Natacyn) was approved on the same day as Candicidin and is the only listed and prescribed Natamycin. Ten out of the 15 entries of Amphotericin B have been discontinued and only five manufacturers are still producing it. Nystatin has 123 entries, with 62 discontinued and 61 approved manufacturers [22].

All these data suggest that the diversity of polyene macrolide antifungal antibiotics remains limited. When equimolar single ingredients of nystatin A1, A3, and polyfungin B were tested, polyfugin B exhibited a better fungicidal activity than nystatin A1 and A3. These promising results warrant further extensive metabolic and biological studies of polyfugin B, nystatin A3, and other structurally similar ingredients in the near future.

**Supplementary Materials:** The following supporting information can be downloaded at https://www.mdpi.com/article/10.3390/catal13020247/s1, Table S1. Orthogonal experimental factors and level assignments. Table S2. Orthogonal experimental results and analysis.

**Author Contributions:** M.S. (data curation, investigation); W.H. (investigation, methodology); S.C. (data curation); F.W. (resources); W.X. (Weizhuo Xu) and W.X. (Wei Xu) (resources, supervision, writing—review and editing). All authors have read and agreed to the published version of the manuscript.

**Funding:** This research received no external funding.

**Data Availability Statement:** Data are available upon reasonable request.

**Conflicts of Interest:** The authors declare no conflict of interest.

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
