# Peer review of "Nysfungin Production Improvement by UV Mutagenesis in Streptomyces noursei D-3-14"

_catalysts, doi:10.3390/catal13020247_

Round 1
Reviewer 1 Report
The paper deals with the nysfungin production improved by the UV irradiation to microbe. This manuscript contains interesting results, so that the paper can be published the journal after the following points are revised.
— The authors should discuss the mechanism of this UV irradiation effect to the production at molecular level.
— Almost table contains long digit number of significant figures. The authors should check the digit number on each table.
Author Response
Dear editor and reviewer,
Thank you for your email and reviewer's comments on our manuscript entitled "Nysfungin production improvement by UV mutagenesis in Streptomyces noursei D-3-14". Our reviewer’s careful reviewing work are highly appreciated.
The responses to reviewer’s comments and the adaptive corrections are as follows:
1.The authors should discuss the mechanism of this UV irradiation effect to the production at molecular level.
Response to comment #1:
The UV irradiation effect had been described in molecular level in detail in lines 59-69.
2.Almost table contains long digit number of significant figures. The authors should check the digit number on each table.
Response to comment #2:
All significant digits in the table have been modified.
Reviewer 2 Report
Comments for catalysts-2164202
In this work, Streptomyces noursei D-3-14 was taken as the starting strain and treated with UV (15 W, 30 cm) for 40 seconds for mutagenesis. After consecutive rounds of screening following the UV mutagenesis, the chemical and biological potencies were increased step by step, from 5783 U/mL and 4626 U/mL (74-14-8) through 8912 U/mL and 5557 U/mL (74-22-1) to 11097 U/mL and 10751 U/mL (112-24-63). Unfortunately, since the mutant 112-24-63 was not genetically stable, the authors gave up this strain. As a whole story, I suggest the authors to screen the stable mutant from the progeny of 112-24-63, since the latter was much higher on the chemical potency and the biological potency.
Other comments:
1) Figure 5 and Table 4: According to the text, the numbers should be 74-22-1,74-22-3…, please check the data of the whole text carefully.
2) L138: Delete the repeated “to the”.
3) Figure 9: Using English characters to replace the Chinese characters.
4) L157: The title should be “Chemicals, strains, …”.
5) Please review the spaces between the number and the unit throughout the document: e.g. 50.0 mL, 24 hr, 3 d, 180 rpm, 30 min, 20 s, 100 mL...
6) Using the correct symbol of degree Celsius (°C).
7) L172: Using the subscript 3 for CaCO3.
8) L239: Add “with” after “supplemented”: were supplemented with…
9) Most of the references are too old and the total number is not enough.
Author Response
Dear editor and reviewer,
Thank you for your email and reviewer's comments on our manuscript entitled "Nysfungin production improvement by UV mutagenesis in Streptomyces noursei D-3-14". Our reviewer’s careful reviewing work are highly appreciated.
The responses to reviewer’s comments and the adaptive corrections are as follows:
In this work, Streptomyces noursei D-3-14 was taken as the starting strain and treated with UV (15 W, 30 cm) for 40 seconds for mutagenesis. After consecutive rounds of screening following the UV mutagenesis, the chemical and biological potencies were increased step by step, from 5783 U/mL and 4626 U/mL (74-14-8) through 8912 U/mL and 5557 U/mL (74-22-1) to 11097 U/mL and 10751 U/mL(112-24-63). Unfortunately, since the mutant 112-24-63 was not genetically stable,the authors gave up this strain. As a whole story, I suggest the authors to screen the stable mutant from the progeny of 112-24-63, since the latter was much higher on the chemical potency and the biological potency.
Response to this comment:
Thanks for our reviewer’s suggestions. This work is being doing in the laboratory.
1.Figure 5 and Table 4: According to the text, the numbers should be 74-22-1,
74-22-3…, please check the data of the whole text carefully.
Response to comment #1:
Thanks for our reviewer’s careful reviewing. These texts had been corrected.
2.L138: Delete the repeated “to the”.
Response to comment #2:
Thanks for our reviewer’s careful reviewing. The repeated parts had been deleted.
3.Figure 9: Using English characters to replace the Chinese characters.
Response to comment #3:
Thanks for our reviewer’s careful reviewing. These Chinese characters had been deleted.
4.L157: The title should be “Chemicals, strains, …”.
Response to comment #4:
Thanks for our reviewer’s careful reviewing. The title had been changed to “Chemicals, strains, culture media and growth conditions” in line 215.
5.Please review the spaces between the number and the unit throughout the
document: e.g. 50.0 mL, 24 hr, 3 d, 180 rpm, 30 min, 20 s, 100 mL...
Response to comment #5:
Thanks for our reviewer’s careful reviewing. These spaces had been unified.
6.Using the correct symbol of degree Celsius (°C).
Response to comment #6:
Thanks for our reviewer’s careful reviewing. The corrected degree Celsius (°C) had been used.
7.L172: Using the subscript 3 for CaCO3.
Response to comment #7:
Thanks for our reviewer’s careful reviewing. The subscript 3 for CaCO3 had been corrected in line 230.
8.L239: Add “with” after “supplemented”: were supplemented with…
Response to comment #8:
Thanks for our reviewer’s careful reviewing. The sentence had been modified in line 298 and 299.
9.Most of the references are too old and the total number is not enough.
Response to comment #9:
Thanks for our reviewer’s careful reviewing. Another 10 references had been supplemented. It is true that there are a few articles on the structure and composition of nystatin, so the relevant literatures are indeed limited.
Reviewer 3 Report
Nystatin is a polyene macrolide antibiotics and has extensive applications. In the manuscript authored by Song et al., Streptomyces noursei was mutated to improve Nysfungin production improvement by UV mutagenesis. The mutants were screened and a strain S. noursei 72-22-1 with higher chemical potency and biological potency were obtained. The mutant was stable and the fermentation was optimized, the mutant strain 72-22-1 displayed a higher content of polyfungin B. The work provides a new strain for polyfungin B production and has application potential.
Here are some comments for consideration:
1: The title needs to be rewritten.
2: The aim of the work was not well described in the introduction part, it should be rewritten and enriched.
3: Figure 4, figure 5, and figure 6 do not provide valuable information, the figures can be removed since tables 3, 4, and 5 provided enough information.
4: Figure 7, the generation passages can be extended further, it’s not enough the draw the conclusion that the mutant obtained was genetically stable.
5: 2. Orthogonal optimization for the nysfungin fermentation and glucose concentration were combined, and the components of the two substrates should be discussed.
6: Table 7. Orthogonal experimental results and analysis are not necessary included in the main context.
Author Response
Dear editor and reviewer,
Thank you for your email and reviewer's comments on our manuscript entitled "Nysfungin production improvement by UV mutagenesis and fermentation optimization in Streptomyces noursei D-3-14". Our reviewer’s careful reviewing work are highly appreciated.
The responses to reviewer’s comments and the adaptive corrections are as follows:
1.The title needs to be rewritten.
Response to comment #1:
Thanks for our reviewer’s careful reviewing. The title had been corrected to “Nysfungin production improvement by UV mutagenesis in Streptomyces noursei D-3-14”.
2.The aim of the work was not well described in the introduction part, it should be
rewritten and enriched.
Response to comment #2:
Thanks for our reviewer’s careful reviewing. The introduction part had been rewritten and enriched.
- Figure 4, figure 5, and figure 6 do not provide valuable information, the figures
can be removed since tables 3, 4, and 5 provided enough information.
Response to comment #3:
Thanks for our reviewer’s careful reviewing. Figures 4-6 had been removed.
4.the generation passages can be extended further, it’s not enough the draw the conclusion that the mutant obtained was genetically stable.
Response to comment #4:
Thanks for our reviewer’s careful reviewing. Normally, a five-generation test was use to evaluate the genetic stability. If it works well, it could be literally called “genetic stable”. To demonstrate the real genetic stability, there are much detailed work would be performed.
5.Orthogonal optimization for the nysfungin fermentation and glucose concentration were combined, and the components of the two substrates should be discussed.
Response to comment #5:
Thanks for our reviewer’s careful reviewing. These contents had been discussed and corrected, and also the manuscript structures had been adjusted.
6.Table 7. Orthogonal experimental results and analysis are not necessary included in the main context.
Response to comment #6:
Thanks for our reviewer’s careful reviewing. These contents had been moved to the supplementary information.